# Antioxidant Effect of Extracts from Native Chilean Plants on the Lipoperoxidation and Protein Oxidation of Bovine Muscle

**DOI:** 10.3390/molecules24183264

**Published:** 2019-09-07

**Authors:** Raquel Bridi, Ady Giordano, Maria Fernanda Peñailillo, Gloria Montenegro

**Affiliations:** 1Departamento de Farmacia, Facultad de Química y de Farmacia, Pontificia Universidad Católica de Chile, Santiago 7820436, Chile; 2Departamento de Química Inorgánica, Facultad de Química y de Farmacia, Pontificia Universidad Católica de Chile, Santiago 7820436, Chile; 3Departamento de Ciencias Vegetales, Facultad de Agronomía e Ingeniería Forestal, Pontificia Universidad Católica de Chile, Santiago 7820436, Chile

**Keywords:** plant extracts, native Chilean species, bovine muscle, meat, lipid oxidation, protein oxidation, antioxidant

## Abstract

The present study investigated the antioxidant potential and the ability to inhibit lipid and protein oxidation in bovine meat of four native Chilean species: canelo (*Drimys winteri*), nalca (*Gunnera tinctoria*), tiaca (*Caldcluvia paniculata*), and ulmo (*Eucryphia cordifolia*). Phenolic acids (gallic, chlorogenic, caffeic, and coumaric) and flavonoids (catechin, epicatechin, and rutin) were identified and quantified by HPLC-MS/MS. *Drimys winteri* extract exhibited the highest antioxidant capacity evaluated by oxygen radical absorption capacity-red pyrogallol method (ORAC-PGR) and ferric ion reducing antioxidant power (FRAP) assays. All extracts decreased lipid oxidation induced by 2,2’-azo-bis(2-amidinopropane) dihydrochloride (AAPH) derived peroxyl radicals by anywhere between 30% and 50% the. In addition, canelo and nalca extracts decreased spontaneous oxidation by around 57% and 37% in relation to the control group, being even more efficient than butylated hydroxyanisole (BHT) a synthetic antioxidant. Protein oxidation in the myofibrillar proteins was evaluated by the formation of protein carbonyls and loss of protein thiols. The canelo, ulmo, and nalca extracts decreased the formation of carbonyls by around 30%. Plant extracts and BHT did not show an antioxidant effect on protein thiol loss. However, tiaca and ulmo extracts exerted a pro-oxidant effect, favoring the oxidation of sulfhydryl groups. The oxidizing system induced structural changes in myofibrillar protein (SDS−PAGE). A protective effect on protein structure from the canelo extract can be observed during the incubation when compared to samples incubated with AAPH.

## 1. Introduction

The preservation of meat products has always been a constant concern in the food industry, in order to maintain the quality and also the safety of the product. To mitigate the oxidative process, the most important mechanism behind the deterioration of meat, the industry has used several molecules with antioxidant capacity. Products of natural origin have been prioritized mainly due to the growing concern among consumers about synthetic antioxidants and their potential adverse effects [1,2]. The oxidation processes can affect both lipids and proteins, and are facilitated by the presence of various oxidizing agents, for example peroxyl radicals (ROO•). Lipid peroxidation is the main cause of deterioration of meat products, causing the appearance of stale odors, changes in the taste and color of the meat, and is partly responsible for protein oxidation. Protein oxidation corresponds to chemical modifications produced in the structure of the protein [3,4,5]. A large number of researches have indicated that complex extracts from plants, such as fruits, vegetables, herbs, and spices were good sources of natural antioxidants in meat and meat products [6,7,8,9,10].

The continental Chilean territory has diverse geomorphology, which together with the biogeographical isolation of a territory limited by geographical and climatic barriers, has given it a biodiversity characterized by a high level of endemism in its ecosystems. The Chilean vascular flora are close to being made up of 50% endemic plant species [11]. Considering the limited knowledge on secondary metabolites and the varied backgrounds on the biological activity regarding these native and endemic plants, these species constitute a large source of bioactive compounds. The native Chilean species canelo, *Drimys winteri* J.R. Forst and G. Forst (Winteraceae); nalca, *Gunnera tinctoria* (Molina) Mirb (Gunneraceae); tiaca, *Caldcluvia paniculata* (Cav.) D. Don (Cunoniaceae) and ulmo, *Eucryphia cordifolia* Cav. (Cunoniaceae) are very emblematic species in this country [12,13,14]. Despite widespread use in folk medicine, these plants have been scarcely explored from a chemical and pharmacological standpoint. A few previous studies reported that these plants contains antioxidant components including flavonoids, tannins, diterpenes, steroids, and volatile oils [15,16,17]. 

In this sense, this work focused on the study of the antioxidant potential and the ability to inhibit lipid and protein oxidation in meat of the following species: *Drimys winteri*, *Gunnera tinctoria*, *Caldcluvia paniculata,* and *Eucryphia cordifolia.* For the chemical characterization of plant extracts, total phenols and flavonoids were quantified and the main phenolic compounds were identified by UHPLC-MS/MS. The antioxidant capacity of plant extracts was evaluated using the oxygen radical absorption capacity-red pyrogallol method (ORAC-PGR) and ferric ion reducing antioxidant power (FRAP) assay. Furthermore, the antioxidant ability of these plant extracts to inhibit oxidative modifications induced by peroxyl radicals (ROO•) in bovine meat were evaluated. Lipid peroxidation was assessed using the TBARS technique and protein oxidation was evaluated by quantification of carbonyl and sulfhydryl groups and by electrophoresis in polyacrylamide gels (SDS-PAGE).

## 2. Results and Discussion

### 2.1. Content of Phenolic Compounds, Flavonoid Compounds, ORAC-PGR, and FRAP Values of Plant Extracts

The results for phenolic content and antioxidant capacity in the plants extracts are presented in Table 1. There is a wide variation in total phenolic content ranging from 91 to 438 mg of gallic acid equivalents (GAE)/100g dry plant weight (DW) and in flavonoid content from 47 to 90 mg of quercetin equivalents (QE)/100g DW. The *D. winteri* (canelo) extract presented a higher content of total phenolic compounds but its flavonoid content is similar to the ulmo extract. The analysis of the antioxidant capacity of the extracts was performed using a direct assay method based on hydrogen atom transfer (ORAC) and an indirect assay based on single electron transfer (FRAP). ORAC-PGR values varied between 3 and 14 µmol equivalents of Trolox (TE)/100g DW whereas FRAP varied between 27 and 95 mg FeSO_4_·7H_2_O /100g DW, being the highest values presented by canelo extract. The results showed a statistically significant positive correlation between total phenolic content (TPC) of samples and the antioxidant capacity measured by FRAP (0.99; *p* ≤ 0.01). Despite the scarce studies with these Chilean native species the presence of phenolic and flavonoid compounds with antioxidant capacity in the *D. winteri* (canelo) and *G. tinctoria* (nalca) are described [15,16,17]. To get more insights about the phenolic composition of the studied samples, we determined the concentration of the most representative polyphenols (cinnamic acids, flavonols, flavone, and flavanone); the results are depicted in Table 2.

### 2.2. Lipid Oxidation in Meat Extract–TBARS 

The TBARS method allows an estimation of the ability of a sample to inhibit malondialdehyde formation resulting from the oxidative degradation of lipids, a process that occurs naturally in meat. The concentration of malondialdehyde (MDA) in meat extract incubated with buffer (control), 2,2’-azo-bis(2-amidinopropane) dihydrochloride (AAPH - ROO• producer), butylated hydroxyanisole (BHT, synthetic chain breaking antioxidant) and plant extracts was measured at 0 and 2 h of incubation. Figure 1 shows the percentage increase of MDA of each sample after 2 h. A 50% increase in the MDA concentration in the control sample can be observed, where at the second hour of incubation (spontaneous oxidation) it went from 0.4 to 0.6 μmol equivalents of MDA / kg of meat. The presence of AAPH increased the concentration of MDA at the second hour up to a concentration of 3.5 µmol equivalents MDA / kg meat. The BHT showed a protective effect against the spontaneous oxidation and against the lipid peroxidation induced by AAPH. The *D. winteri* (canelo) and *G. tinctoria* (nalca) extracts showed a significant decrease (*p* < 0.05) 57% and 37% in relation to the control group in spontaneous oxidation, even more efficient than BHT. In addition, all extracts significantly decreased lipid oxidation caused by AAPH, finding percentages decreased by 30.8% (*C. paniculata*), 42.3% (*D. winteri*), 53.8% (*G. tinctoria*), and 54.6% (*E. cordifolia*) calculated in relationship to the AAPH group.

Four phenolic acids (gallic acid, chlorogenic acid, caffeic acid, and coumaric acid) and three flavonoids (catechin, epicatechin, and rutin) were found in all extract samples. The extracts showed a considerable content of catechin and epicatechin mainly the *C. paniculata* (tiaca) extract that presented 519.4 mg/kg DW of catechin and 491.4 mg/kg DW of epicatechin. These two flavonols are widely distributed in the plant species and have been successfully used in preventing lipid oxidation in foods [18,19,20].

### 2.3. Protein Oxidation

#### 2.3.1. Determination of Carbonyl Content in Myofibrillar Proteins

The formation of carbonyl groups have been widely employed as a general index of protein oxidation in meat and derived products [3,21,22]. Considering that the formation of carbonyls involves irreversible modifications of essential amino acids, their accumulation is a clear expression of the detrimental impact on the nutritional value of food proteins [3]. The carbonyl content in myofibrillar proteins incubated with buffer (control), AAPH (ROO• producer), BHT (synthetic chain breaking antioxidant) and plant extracts was measured at 0, 2, 4, and 6 h after incubation. Figure 2 shows the percentages of the increase of carbonyl content of each sample in comparison with its initial carbonyl content.

The initial value of the carbonyl content in the control sample at 0 h was 8.2 ± 0.3 nmol carbonyls/mg of protein reaching up to 10.6 ± 0.08 nmol/mg protein at the sixth hour. The AAPH group reached 15.8 ± 0.1 nmol/mg protein in the sixth hour. The extract of *E. cordifolia* (ulmo) and *D. winteri* (canelo) decreased the formation of carbonyls in the samples treated jointly with AAPH in the second, fourth, and sixth hours of incubation. *G. tinctoria* (nalca) demonstrated protection against oxidation mediated by peroxyl radicals from 4 h of incubation on. In addition, the ulmo extract prevented spontaneous oxidation and exhibited greater efficiency than BHT. On the other hand, *C. paniculata* (tiaca) extract significantly increased the formation of carbonyl groups at all hours in relation to the 10 mM AAPH group, suggesting that the tiaca extract would be acting as a pro-oxidant in this system. Some phenolic-rich plant and fruits extracts have shown protein carbonylation inhibition in chicken meat, cooked patties, frankfurters, and liver pâtés [21]. Nevertheless, the effect of the plant phenolics against protein oxidation is governed by specific interaction mechanisms between the phenolic compounds and the proteins. These interactions are dependent on the amount and chemical state of the phenolic compound and the size, structural conformation, and overall charge of the protein. According to Estévez and Heinonem [23], certain phenolic compounds (i.e. gallic acid, catechin, cyanidin-3-glucoside, and rutin) would be able to inhibit the formation of specific protein carbonyls from myofibrilar proteins. On the other hand, some phenolic compounds such as chlorogenic acid and catechin exhibited both antioxidant and pro-oxidant effects. These inconsistent effects of phenolic compounds may be attributed to their concentration, the oxidative conditions and the target in proteins. *C. paniculata* (tiaca) extract showed a considerable content of catechin and epicatechin (Table 2). The pro-oxidant activity is attributed to autoxidation or peroxidase-based oxidation and generation of reactive oxygen species (ROS) including phenoxyl radicals or the quinone-dependent oxidation of low molecular weight antioxidants [23,24,25].

#### 2.3.2. Determination of Total Sulfhydryl Groups in Myofibrillar Proteins

The loss of sulphydryl groups has been used as a marker of protein oxidation in meat products since the content of thiols in meat is not affected by post-mortem aging and the cysteine residues show high susceptibility to oxidation [22]. Figure 3 showed the content of thiols in myofibrillar proteins incubated with buffer (control), AAPH (ROO• producer), BHT (synthetic chain breaking antioxidant), and plant extracts at 0, 2, 4, and 6 h of incubation.

Sulphydryl contents decreased during incubation and the initial value in the control sample at the beginning was 196.7 nmol/mg protein and it showed a decrease of about 52% by the sixth hour. The AAPH group showed a decrease, near 63%, in the sixth hour. These values are not significantly different, indicating 10 mM AAPH is not efficient in increasing the oxidation of sulfhydryl groups. Previous studies in plasma proteins showed that the concentration of sulfhydryl groups does not change in the presence of peroxyl radicals, suggesting that reduced thiol groups of proteins can inhibit the formation of hydroperoxides by eliminating peroxyl radicals [26]. Plant extracts and BHT did not show protection against oxidation. Tiaca extract (*C. paniculata*) showed an increase in oxidation when compared to the control group, at the second, fourth and sixth hours, suggesting that this extract exerts a pro-oxidant effect, favoring the oxidation of sulfhydryl groups. The same behavior was also observed for the ulmo extract (*E. cordifolia*) at the second and fourth hours of treatment. A study using green tea (*Camellia sinensis*), a commonly used antioxidant extract in various food products, and maté (*Ilex paraguariensis*) showed a decreased in the thiol concentration when added to pork loins, indicating a pro-oxidative activity [27].

#### 2.3.3. SDS-PAGE Electrophoresis

Electrophoresis was performed in order to observe modifications induced by protein oxidation. In the gels (Figure 4), bands corresponding to the different proteins present in the extract are observed. Actin and myosin stand out as the major component of the extract, with a molecular weight (MW) of around 42 kDa and 200 kDa, respectively. 

Figure 4A shows the SDS−PAGE of myofibrillar protein after 0, 2, 4, and 6 h of incubation with buffer or AAPH-derived radicals. It can be observed that the oxidizing system applied had an effect on protein structure from the second hour of incubation, which is similar to that observed in literature [4]. No changes were observed in the structure of the protein in the controls at any hour of incubation. Figure 4B shows that the protein bands incubated with *D. winteri* (canelo) (left side) do not suffer any modification in the incubation periods (0, 2, 4, and 6 h). Also, a slight protection of the extract, visible mainly in actin, can be observed during the hours of incubation compared to the samples incubated with AAPH (4A). The electrophoresis carried out for the extracts of *G. tinctoria* (nalca), *C. paniculata* (tiaca), and *E. cordifolia* (ulmo) did not show protection in the presence of the oxidizing medium (data not shown). 

## 3. Materials and Methods 

### 3.1. Plant Material

Aerial parts of *Drimys winteri* (canelo), *Gunnera tinctoria* (nalca), *Caldcluvia paniculata* (tiaca) and *Eucryphia cordifolia* (ulmo) were collected in Región de los Ríos, Región de los Lagos and Chiloé. The species were identified by Miguel Gomez (UC, Santiago, Chile). Vouchers were deposited at the Herbarium of Facultad de Agronomía de la Pontificia Universidad Católica de Chile.

### 3.2. Plant Extracts

The plant material was dried at room temperature and powdered. The extracts were obtained by kinetic maceration (24 h) employing 70% ethanol at room temperature. The extract was filtered (0.45 um) and the solvent was evaporated to dryness under reduced pressure (45 °C). The dry extract was reconstituted in distilled water (2 g/mL), refiltered and stored at −80 °C in the dark until further use.

### 3.3. Total Phenolic Content (TPC) 

A diluted solution of plant extracts in distilled water was mixed with 2.5 mL of the Folin Ciocalteu reagent 1:10 (*v/v*) and 2.0 mL of a solution of Na_2_CO_3_. After 60 min, the absorbance of the resulting blue solution was measured at 760 nm using an Agilent 8453 UV-visible spectrophotometer (Palo Alto, CA, U.S.A.). Results are expressed as milligrams of gallic acid equivalents per 100 g of dried weight (mg GAE/100g DW). Values are reported as mean ± standard deviations (SD) of three replicates in triplicate [28].

### 3.4. Flavonoids Determination

A diluted solution of plant extracts in distilled water was mixed with 0.5 mL of AlCl_3_ 2% (*w/v*) in methanol solution. After 60 min at room temperature, the absorbance was measured at 420 nm using an Agilent 8453 UV-visible spectrophotometer (Palo Alto, CA, U.S.A.). Total flavonoid contents were calculated as milligrams of quercetin equivalents per 100 g of dried weight (mg QE/100g DW). Values are reported as mean ± SD of three replicates in triplicate [28].

### 3.5. Identification and Quantification of Flavonoids and Phenolic Acids

An ABSciex triple Quad 4500 mass spectrometer equipped with an electrospray (TurboV) interface coupled to an Eksigent Ekspert Ultra LC100 with Ekspert Ultra LC100-XL autosampler system (AB/Sciex Concord, Ontario, ON, Canada) was used. Separation of analytes was achieved at 30 °C on a LiChrospher 100 RP-18 end-capped column (125 mm × 4 mm id, 5 µm) (Merck, Darmstadt, Germany) with a gradient elution of 0.1% formic acid in water (A) and methanol (B) as the mobile phase, with a gradient as follows: 0–1 min, 15% B; 1–17 min, 15%–100% B; 17–21 min 100% B; 21–22 min, 100%–15% B; 22–25 min, 15% B, while the flow rate was kept at 0.5 mL/min and the injection volume was 10 µL. Quantification was performed with calibration curves using commercially available standards.

### 3.6. Determination of Antioxidant Capacity

The ferric ion reducing antioxidant power (FRAP) and oxygen radical absorbance capacity (ORAC-PGR) methods were used as described by Velasquez et al. [29] and Bridi et al. [28], respectively. The FRAP results were expressed as mg FeSO_4_·7H_2_O/100g DW and ORAC as µmol TE/100g DW. 

### 3.7. Lipid Oxidation 

#### 3.7.1. Preparation and Incubation of Meat Extract

Five grams of fresh bovine muscle (*Longissimus dorsi*) was homogenized in 10 mL of milliQ water, using an Ultra-Turrax homogenizer at 11,000 rpm for 2 min in an iced water bath. The homogenates were incubated with 7.5 mM phosphate buffer at pH 7.4 (negative control), butylated hydroxyanisole (BHT) 1 mM (positive control) or plant extracts (0.1 g/mL). In addition, aqueous ROO• producer AAPH (2,2′-azobis-(2-amidinopropane) hydrochloride) 10 mM or buffer was added to each sample to complete the final volume. The incubation was done in the dark at 37 °C for 2 h and air was bubbled each hour in order to maintain oxygen availability [4]. Aliquots were taken at time 0 and 2 h, where 0 h was considered the time immediately after all components of the incubation were mixed. To prevent subsequent oxidation, Trolox 0.1 mM (water-soluble α-tocopherol analogue) was added and immediately cooled to 0–4 °C by ice-bathing and used for analysis. 

#### 3.7.2. TBARS Determination

Lipid oxidation was measured by TBARS formation according to the method of Esterbauer and Cheeseman [30]. Meat extracts samples (2 mL) were mixed with 1mL 10% (*w*/*v*) trichloroacetic acid, centrifuged at 3857× *g* for 15 min at 4 °C, and the supernatant was filtered (0.45 µm). 500 µL were transferred to a Pyrex tube with 500 µL of thiobarbituric acid TBA (20 mM). The mixture was homogenized, placed in a boiling water bath for 15 min, and subsequently cooled. Absorbance was measured at 532 nm in a spectrophotometer (Agilent 8453 UV-visible Spectrophotometer) and a calibration curve using 1,1,3,3-tetramethoxypropane was performed while being subjected to the same treatment as that of the samples. AAPH, BHT, and the plants extracts did not produce color when tested without the addition of the supernatant, demonstrating the absence of a direct reaction to thiobarbituric acid. TBARS values were calculated as µmol equivalents of malondialdehyde (MDA)/kg meat and represented as percentage of initial control concentrations (mean ± SD of 3 replicates in triplicate)

### 3.8. Protein Oxidation

#### 3.8.1. Preparation and Incubation of Myofibrillar Proteins

Myofibrillar proteins were prepared according to the method of Martinaud et al. [31] and modified by Zhou et al. [4]. Briefly, the fresh bovine muscle (*Longissimus dorsi*) was homogenized in 100 mL of a sodium phosphate buffer solution (20 mM) at pH 6.5 containing 150 mM NaCl, 25 mM KCl, 3 mM MgCl2, and 4 mM EDTA, using an Ultra-Turrax homogenizer at 11,000 rpm for 1 min in an iced water bath. The homogenate was centrifuged at 2000× *g* for 15 min at 4 °C. The pellet was washed twice with 100 mL of a 50 mM KCl solution at pH 6.4 and once with 100 mL of 20 mM sodium phosphate buffer at pH 6.0. The pellet was finally suspended in 20 mM phosphate buffer containing 0.6 M NaCl (pH 6.0) and the protein concentration was adjusted to calculated concentrations by the method bicinchoninic acid (BCA) assay using BSA as a standard [32]. Myofibrillar protein (0.7–1.0 mg/mL) was incubated with 7.5 mM phosphate buffer at pH 7.4 (negative control), BHT 1 mM (positive control) or plant extracts (0.1g/mL). In addition, aqueous ROO• producer AAPH 10 mM or buffer was added to each sample to complete the final volume. The incubation was done in the dark at 37 °C for 2 h and air was bubbled every hour in order to maintain oxygen availability [4]. Aliquots were taken at time 0, 2, 4 and 6 h. To prevent subsequent oxidation, Trolox 0.1 mM (water-soluble α-tocopherol analogue) was added and immediately cooled to 0–4 °C by ice-bathing and used for analysis. 

#### 3.8.2. Determination of Carbonyl Content

Carbonyl groups were detected by their reaction with 2,4 dinitrophenylhydrazine (DNPH) to form protein hydrazones, and their content was estimated using the method of Levine et al. [33] with modifications by Lund et al. [34]. Absorbance was measured at 370 nm in a Biotek synergy HT Multilector, and the blank value (6 M guanidine-HCl) was subtracted from the corresponding sample value. The amount of carbonyl was expressed as nmol equivalents/mg of protein using an absorption coefficient of 22,000 M^−1^cm^−1^ for protein hydrazones. The results are presented as percentage of controls (mean ± SD of three replicates in triplicate). 

#### 3.8.3. Determination of Total Sulfhydryl Groups

The sulfhydryl group levels were determined by the method originally described by Elmman [35] and modified by Sun et al. [36]. Thiols interact with 5, 5’-dithiobis-(2-nitrobenzoic acid) (DTNB), forming a highly colored anion with maximum peak at 412 nm (absorption coefficient of 13,600 M^−1^cm^−1^). Absorbance was measured in an Agilent 8453 UV-visible Spectrophotometer and the concentration of sulfhydryl groups were expressed in µmoles/mg of protein. The results are presented as percentage of controls (mean ± SD of three replicates in triplicate). 

#### 3.8.4. SDS-PAGE Electrophoresis

Protein modifications were examined by SDS–PAGE electrophoresis [4]. Incubated samples were mixed and boiled for 5 min in a 62.5 mM Tris buffer (pH 6.8) solution containing 2% sodium dodecylsulfate (SDS), 10% glycerol, 100 mM β-mercaptoethanol and traces of bromophenol blue. Acrylamide (3%) stacking gel, 12% and 8% acrylamide resolving gels, and a running buffer comprising 25 mM Tris, 400 mM glycine, and 0.1% SDS, pH 8.3, were used. Electrophoresis was performed at 100 V over 1−2 h. Gels were stained with 0.1% Colloidal Coomassie and distained in water over 48 h. Gels were scanned and performed using ImageJ software. Broad range protein standards were obtained from Bio-Rad Laboratories (Bio-Rad Laboratories Inc., Herts, UK).

### 3.9. Statistical Analysis

Statistical analyses were performed using Pearson parametric correlations and one-way analysis of variance (ANOVA) followed by the Tukey when the F value was significant. All analyses were carried out using Origin Pro 8 software. A value of *p* < 0.05 was considered significant.

## 4. Conclusions

This study demonstrated the ability to inhibit lipid and protein oxidation in meat of the canelo (*Drimys winteri*), nalca (*Gunnera tinctoria*), tiaca (*Caldcluvia paniculata*), and ulmo (*Eucryphia cordifolia*) species. All extracts significantly decreased lipid oxidation caused by AAPH, while canelo and nalca extracts showed a significant decrease in relation to the control group in spontaneous oxidation, even more efficient than BHT. *E. cordifolia* (ulmo), *D. winteri* (canelo), and *G. tinctoria* (nalca) extracts decreased the formation of carbonyls in the samples treated jointly with AAPH demonstrating protection against oxidation mediated by peroxyl radicals. On the other hand, the *C. paniculata* extract significantly increased the formation of carbonyl groups, suggesting that the tiaca extract could be acting as a pro-oxidant in this system. A protective effect on protein structure (SDS−PAGE) from the canelo extract can be observed during the incubation when compared to samples incubated with AAPH. Overall, canelo extract showed the best results in terms of protection against lipid and protein oxidation in bovine meat. All results suggested that phytochemicals from native Chilean plants could be promising as active ingredients for antioxidant purposes.

## Figures and Tables

**Figure 1 molecules-24-03264-f001:**
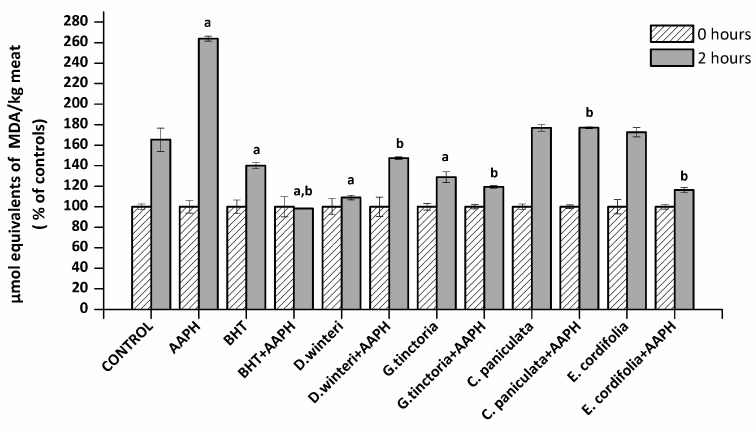
Concentration of malondialdehyde (MDA) in the different samples analyzed, control, 2,2’-azo-bis(2-amidinopropane) dihydrochloride (AAPH, 10 mM), butylated hydroxyanisole (BHT) (1 mM) and plant extracts (0.1 g/mL), (*p* < 0.05; ^a^ significant difference compared to control group; ^b^ significant difference compared to AAPH group, ANOVA, Tukey test). Data are expressed as percentages of initial concentration and represent the mean ± SD for of three replicates in triplicate.

**Figure 2 molecules-24-03264-f002:**
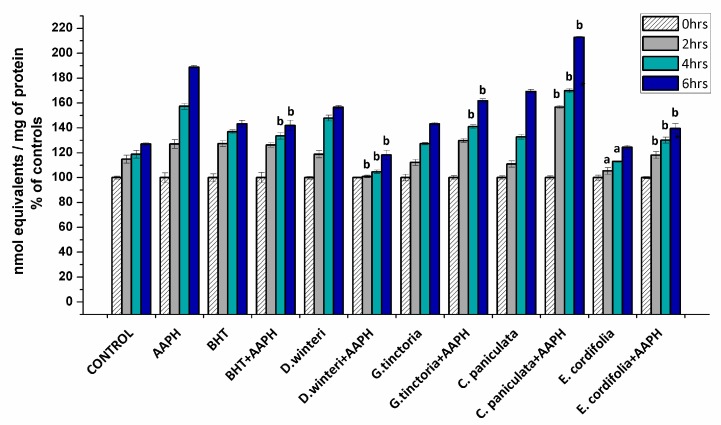
Concentration of carbonyl groups in the different samples analyzed, control, AAPH (10 mM), BHT (1 mM), and plant extracts (0.1 g/mL), (*p* < 0.05; ^a^ significant difference compared to control group; ^b^ significant difference compared to AAPH group, ANOVA, Tukey test). Data are expressed as percentage of initial concentrations and represent the mean ± SD for of three replicates in triplicate.

**Figure 3 molecules-24-03264-f003:**
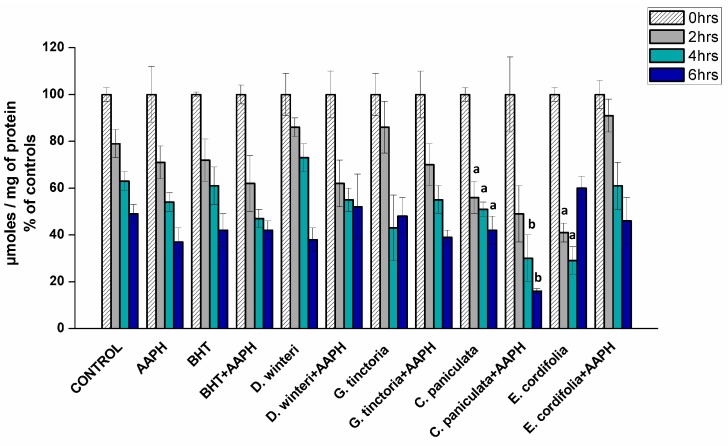
Content of thiols in the different samples analyzed, control, AAPH (10 mM), BHT (1 mM), and plant extracts (0.1 g/mL), (*p* < 0.05; ^a^ significant difference compared to control group; ^b^ significant difference compared to AAPH group, ANOVA, Tukey test). Data are expressed as percentage initial concentrations and represent the mean ± SD for of three replicates in triplicate.

**Figure 4 molecules-24-03264-f004:**
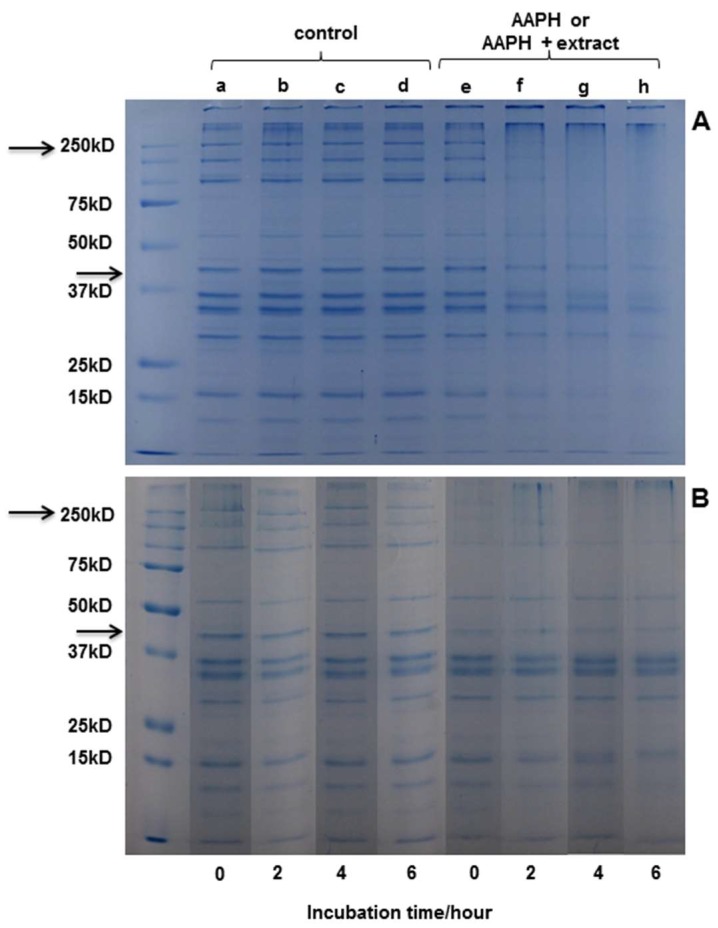
SDS-PAGE analysis of myofibrillar proteins incubated with different systems. (**A**) Buffer (a, b, c, and d) or peroxyl radicals (e, f, g, and h) generated from 10 mM AAPH in 100 mM phosphate buffer, pH 7.4, at 37 °C for 0, 2, 4, or 6 h. (**B**) Buffer (a, b, c, and d) or *D. winteri* (canelo) plus peroxyl radicals (e, f, g, and h) generated from 10 mM AAPH in 100 mM phosphate buffer, pH 7.4, at 37 °C for 0, 2, 4, or 6 h. (MW 250 kD corresponding to myosin; MW 47 kD corresponding to actine).

**Table 1 molecules-24-03264-t001:** Content of phenolic and flavonoid compounds, oxygen radical absorption capacity-red pyrogallol method (ORAC-PGR) and ferric ion reducing antioxidant power (FRAP) values in plant extracts.

Plants	Total Phenolic Content (mg GAE/100g DW)	Flavonoid Content (mg QE/100g DW)	ORAC-PGR (µmol TE/100g DW)	FRAP (mg FeSO_4_·7H_2_O /100g DW)
*D. winteri* (canelo)	438 ± 16 ^a^	90 ± 0.8 ^a^	14 ± 0.4 ^a^	95 ± 1.1 ^a^
*G. tinctoria* (nalca)	211 ± 15 ^b^	64 ± 0.1 ^b^	5 ± 1.2 ^b^	55 ± 0.9 ^b^
*C. paniculata* (tiaca)	231 ± 3 ^b^	47 ± 0.3 ^c^	3 ± 0.9 ^b^	61 ± 1.6 ^c^
*E. cordifolia* (ulmo)	91 ± 1 ^c^	89 ± 0.7 ^a^	5 ± 1.1 ^b^	27 ± 1.1 ^d^

^a,b,c,d^ Means within each column with different superscripts are significantly different (*p* < 0.05; ANOVA, Tukey test).

**Table 2 molecules-24-03264-t002:** Phenolic composition in plant extracts by UHPLC-MS/MS.

	Concentration (mg / kg DW) ^1^			
	*D. winteri* (canelo)	*G. tinctoria* (nalca)	*C. paniculata* (tiaca)	*E. cordifolia* (ulmo)
Gallic acid	3.88 ± 0.39	2.96 ± 0.24	0.62 ± 0.06	9.03 ± 0.33
Chlorogenic acid	0.81 ± 0.05	ND	1.13 ± 0.07	ND
Caffeic acid	0.20 ± 0.01	0.12 ± 0.01	0.28 ± 0.02	4.15 ± 0.68
Coumaric acid	0.84 ± 0.03	0.12 ± 0.01	0.80 ± 0.03	1.37 ± 0.20
Catechin	52.1 ± 10.2	188.4 ± 15.1	519.4 ± 22.3	0.30 ± 0.03
Pinocembrin	0.06 ± 0.005	ND	ND	0.06 ± 0.01
Rutin	1.33 ± 0.13	5.47 ± 0.33	1.36 ± 0.03	40.4 ± 4.0
Chrysin	0.07 ± 0.01	ND	ND	ND
Quercetin	2.98 ± 0.18	ND	10.15 ± 0.61	7.37 ± 0.67
Abscisic acid	ND	0.09 ± 0.01	0.08 ± 0.05	0.10 ± 0.01
Luteolin	0.30 ± 0.03	ND	ND	ND
Epicatechin	63.66 ± 1.14	197.0 ± 11.8	491.4 ± 3.6	0.54 ± 0.04
Apigenin	0.26 ± 0.01	0.24 ± 0.02	0.30 ±0.01	ND

^1^ mg/kg DW (mg polyphenol/kg dry weight plant).

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
