# Peer review of "Antioxidant Effect of Extracts from Native Chilean Plants on the Lipoperoxidation and Protein Oxidation of Bovine Muscle"

_molecules, 2019, doi:10.3390/molecules24183264_

Round 1
Reviewer 1 Report
The present manuscript describes the antioxidant effects of different plant extracts. The use of plant extracts rich in polyphenols in order to prevent food oxidation has been widely studied in numerous papers; however, the studies on the plants used in the present paper are scarce. The manuscript is well written, the title is suitable for the manuscript (maybe you could change the words “bovine meat” because the study has not been performed in a whole meat).
Specific comments:
-Abstract section:
Presenting data in the abstract would improve its value.
L20-21, This sentence is not supported by a statistical analysis, or it is missing in text or table 1.
L29-30, slight protection? What do you mean? Please explain better.
- Introduction section:
The introduction is appropriate.
L65-66, Please explain the interest of evaluate the modifications induced by peroxyl radical in the meat extract? You homogenize the meat, and incubate at 37 ºC, this is a pro-oxidative scenario per se.
- Results and discussion:
Table 1, please indicate in title that it refers to plant extracts. You should indicate the results of the statistical analysis (significance of the model, the differences among means).
L 75-76, Please check the ranges described in the sentence, it appears to be wrong.
L90-95, In table 2, the tiaca extract presents the highest values of catechin and epicatechin, however in table 1 tiaca extract has the lowest value of flavonoid content. What is the possible explication for this?
L 97-187, In my opinion, it would be better to express the results of TBARS, carbonyl groups and total sulfhydryl groups as concentration (μmol eq MDA/kg meat, nmol equivalents/mg of protein and μmol /mg of protein, respectively) than as the percentage of initial control concentrations.
- Material and methods:
L221, please add space between the number and the units, here and hereafter.
L225 and 232, Commercial brand, city and country should be addressed in the instrumental analysis equipment.
L253, Can you provide the composition of bovine meat? How many extracts did you prepare?
L256, Please change “7,5mM” by “7.5 mM”.
L265-299, As commented above it would be better to express the results of lipid and protein oxidation as concentration than as the percentage of initial concentrations.
L309-314, check the text; it seems that there are some mistakes.
L326, Section should be properly numbered (3.9). Please include the significance level (p).
- Conclusions:
Too long for conclusion. It seems a summary of this study. Conclusion should be concise with the content of implementation after finding of the present study and future works if needed.
Author Response
"Please see the attachment."

Reviewer 2 Report
The manuscript entitled: ‘Antioxidant effect of extracts from native Chilean plants on the lipoperoxidation and protein oxidation of bovine meat presents’ investigates the ability of four native Chilean plant species to inhibit lipid and protein oxidation in bovine meat. The antioxidant capacity, the total phenolic content and flavonoid content of the extracts have been determined. Electrophoresis was performed for the observation of potential modifications induced by protein oxidation. In general, the manuscript is well structured and prepared. Please find my comments below:
Minor comments
Materials and methods
Please write ‘Total phenolic content (TPC) determination’ instead of ‘Total Phenolics Determination (TPC)’
Line 231: 2% w/v? or v/v?
Lines 236-244: Could you please report the column temperature and injection volume?
FRAP results were expressed.
Major
3.1. Plant extracts
Could you please confirm if the extraction protocol used was obtained from the literature? If yes please add the citation. Also, It is not clear what method was used for the dehydration of plant extracts. Freeze drying hot air drying?
Table 1 and Table 2. If the statistical analysis was conducted could you please put letters next to the values?
Instead of antioxidant activity please write antioxidant capacity.
In the 2.1 section discussion should be added. Are the values reported close to those have been reported in the literature before if there are any?
Author Response
"Please see the attachment."

Round 2
Reviewer 2 Report
Authors effectively replied to all the comments made by this reviewer.